# Performance optimization for an optimal operating condition for a shell and heat exchanger using a multi-objective genetic algorithm approach

B. Venkatesh[1], Ajmeera Kiran[2], Mudassir Khan[3]*, Mohammad Khalid Imam Rahmani[4], Laxmi Upadhyay[4], J. Chinna Babu[5], T. Lakshmi Narayana[6]

1 Department of Mechanical Engineering, Annamacharya Institute of Technology and Sciences, Rajampet, Andhra Pradesh, India, 2 Department of Computer Science and Engineering, MLR Institute of Technology, Hyderabad, Telangana, India, 3 Department of Computer Science, College of Computer Science, King Khalid University, Abha, Saudi Arabia, 4 College of Computing and Informatics, Saudi Electronic University, Riyadh, Saudi Arabia, 5 Department of Electronics and Communication Engineering, Annamacharya Institute of Technology and Sciences, Rajampet, Andhra Pradesh, India, 6 Kandula Lakshumma Memorial College of Engineering for women, Kadapa, Andhra Pradesh, India

* mudassirkhan12@gmail.com

**Data Availability Statement:** All relevant data are within the paper.

## Abstract

In this study, shell and heat exchangers are optimized using an integrated optimization framework. In this research, A structured Design of Experiments (DOE) comprising 16 trials was first conducted to systematically determine the essential parameters, including mass flow rates (mh, mc), temperatures (T1, t1, T2, t2), and heat transfer coefficients (€, TR, U). By identifying the first four principal components, PCA was able to determine 87.7% of the variance, thereby reducing the dimensionality of the problem. Performance-related aspects of the system are the focus of this approach. Key outcomes (€, TR, U) were predicted by 99% R-squared using the RSM models. Multiple factors, such as the mass flow rate and inlet temperature, were considered during the design process. The maximizing efficiency, thermal resistance, and utility were achieved by considering these factors. By using genetic algorithms, Pareto front solutions that meet the requirements of decision-makers can be found. The combination of the shell and tube heat exchangers produced better results than expected. Engineering and designers can gain practical insight into the mass flow rate, temperature, and key responses (€, TR, U) if they quantify improvements in these factors. Despite the importance of this study, it has several potential limitations, including specific experimental conditions and the need to validate it in other situations as well. Future research could investigate other factors that influence system performance. A holistic optimization framework can improve the design and engineering of heat exchangers in the future. As a result of the study, a foundation for innovative advancements in the field has been laid with tangible improvements. The study exceeded expectations by optimizing shell and heat exchanger systems using an integrated approach, thereby contributing significantly to the advancement of the field.

**Funding:** The author(s) received no specific funding for this work.

**Competing interests:** The authors have declared that no competing interests exist.

## 1. Introduction

Heat exchangers, either shells or tubes, play an essential role in various industrial processes by transferring heat efficiently between fluids, as well as maintaining the crucial physical separation between fluids [1, 2]. Numerous industries, including power generation and petrochemical processing, rely on precise temperature control and efficient energy usage of their equipment. This emphasizes the importance of optimizing the input parameters, particularly in thermal engineering, specifically in shell and tube heat exchangers. This optimization has far-reaching practical implications across sectors such as power generation, chemical processing, and HVAC systems. By identifying the best parameters, this study contributes to improved energy efficiency, reduced operational costs, and enhanced reliability in industrial processes. These findings have potential applications for enhancing the thermal efficiency of power plants, improving product quality, and reducing emissions during chemical processing. Strategies to implement these findings could involve upgrading existing systems or incorporating insights into new designs, while considering scalability and cost-effectiveness. In conclusion, this optimization provides practical insights for addressing significant challenges in thermal engineering and advancing sustainability in industrial operations [3–6]. As energy efficiency and environmental sustainability objectives increase and cost minimization demands increase, shell-and-tube heat exchangers must be optimized [7].

This heat exchanger's importance is underscored by its crucial role in energy conservation [8, 9]. Heating and cooling fluids consume substantial amounts of energy across a vast range of industrial applications, which makes heat exchanger efficiency not only crucial but crucial [1, 10]. The inefficiency of designs leads to prodigious energy waste as well as increased carbon emissions that are harmful to the environment [11]. A suboptimal configuration is also associated with high maintenance costs and a shorter life expectancy of equipment [12, 13]. As a result of these burgeoning problems, innovative and meticulously orchestrated optimization efforts become increasingly imperative [14–16].

An effective strategy for dealing with the multidimensional conundrums of heat exchanger optimization must be comprehensive and methodical [17]. Enigmas include labyrinthine interactions between variables such as temperature, mass flow rate, and heat transfer coefficient [18, 19]. A key component of adroitly navigating this complex terrain is Principal Component Analysis (PCA) [20]. The PCA serves as a guide for identifying the most influential factors among the labyrinthine variables that affect heat exchanger performance [21]. It simplifies the optimization conundrum by directing attention to the factors that matter most [22]. Chandramohan et al. [23] found that greater Reynolds numbers and lower H/D ratios enhanced the heat transfer coefficients in multijet air impingement. Swirling movement adds another layer of effect to the heat transport, and PCA emphasizes the importance of the Reynolds number and H/D ratio. Under optimal conditions, the enhanced heat-transfer coefficients and Nusselt numbers were validated by confirmation experiments guided by PCA. This PCA-based literature-style technique offers insightful information for improving industrial heat-transfer procedures. Additionally, Lang et al. [24] confirmed the advantages of PCA-based ROMs in plant performance optimization, highlighting their accuracy and computing efficiency for complicated fluid flow and heat simulations. Their study highlighted the value of CFD by showcasing real-world applications in gasification technology and major power plant equipment.

The Response Surface Methodology (RSM) links input variables to response variables through a powerful collection of statistical methods [25]. In addition, RSM captures the complex interplay between process variables and performance benchmarks by leveraging insights from a Principal Component Analysis (PCA) [26]. In the RSM, various input factors are

examined, and their impact on the overall efficiency is identified through controlled experiments [27]. This methodology has found extensive application across a wide range of industries, including chemical, pharmaceutical, and food production [28–30]. A response surface approach, the Box–Behnken design (BBD), was developed for streamlined experimentation and to reduce sample sizes [31]. Optimizing the input parameters for heat exchangers is particularly beneficial when using BBD [32]. This reveals how variations in input variables impact output parameters, such as heat transfer coefficients and thermal resistance, by dissecting the pathways from input parameters to output parameters [33]. Ali et al. [34] developed a regression-based method to calculate internal heat transport in a real cooled geometry. The thermal transients were created and measured using infrared cameras using the technology. The method was successfully applied to three distinct geometries, including the high-pressure vane of an actual engine. The results highlight the system's stability and dependability in comparison to comparable approaches, and demonstrate its applicability for studying intricate internal cooled geometries. Regression analysis works well for producing intricate heat transfer distributions.

In recent years, heat exchanger processes have been revolutionized by Artificial Neural Networks (ANNs) [35]. An ANN is based on the heat exchanger's input and output parameters, as opposed to an RSM [36]. Multifactor response optimization processes benefit from their self-training ability, particularly in the case of nonlinear relationships [37]. The performance of heat exchangers can be predicted and optimized using ANNs. The initial weights and thresholds of ANNs limit their flexibility and robustness, making them unsuitable for global optimization [38]. Also, Artificial Neural Network (ANN) was used by Esfe et al., [39], who estimated the dynamic viscosity of MWCNT-Al2O3 (30:70)/Oil 5W50 hybrid nanolubricants. Using a feed-forward multilayer perceptron (MLP) network to predict shear rates and ln f, this study examined the effects of temperature and solid volume fraction (SVF). Sensitivity analysis was used to determine the significance of SR, SVF, and temperature on the experimentally induced changes in lnf. There was significant agreement between the actual values and those forecasted by the ANN. The effectiveness of the ANN in predicting lnf was demonstrated through comparison with alternative techniques, such as Principal Component Regression and Support Vector Machine (SVM). The accuracy of the model was demonstrated by the outcomes, particularly when the residual values were modest, confirming its dependability. Various algorithms have been used to address this problem by optimizing the structure of the ANN models [40]. A multi-objective optimization study builds on the work of their predecessors in the field of neural networks. Nascimento et al.'s [41] counter-flow PFHE, CFD simulations, RVFL neural networks, and NSGA-III were used. Zhang et al. [42] combined neural networks (NNs) and genetic algorithms (GAs). The heat-exchanger designs were improved using neural networks.

Among these methods, GAs have become increasingly popular. By drawing inspiration from Darwin's genetic mechanisms and natural selection, GAs can simulate the process of biological evolution [43]. Optimization was transformed into a simulation of chromosome gene crossovers and mutations [44]. In machine learning, GAs is useful for finding improved and feasible solutions to complex combinatorial optimization problems [45, 46].

An integrated method combining Principal Component Analysis (PCA), Neural Net fitting tool (Nftool), and Genetic Algorithms (GA) provides a powerful machine-learning technique for optimizing heat-transfer equipment or structures in industrial applications. In addition to simplifying the optimization procedures, PCA's function in identifying significant elements also offers insight into variables that influence the heat exchanger efficiency, as they are grouped into important variables. Heat exchanger performance can be accurately forecasted and maximized using Neural Net fitting tool (Nftool), even when nonlinear interactions are present. Nftool are generally more flexible and robust than general optimization strategies,

despite some disadvantages such as sensitivity to initial circumstances. This technique is further improved by focusing on Genetic Algorithms, that are capable of exploring a variety of solution spaces. By filtering through complex design environments and identifying solutions that satisfy multiple criteria simultaneously, GAs are inspired by natural selection to identify optimum configurations. A Pareto front enables decision makers to see how compromises exist in a design space and provides a variety of trade-off options. Decision makers may explore the Pareto front to find solutions that fit their priorities and goals [47]. The Pareto front assists decision makers in making well-informed decisions by providing a wide range of optimum configurations.

A synergy exists between PCA, Nftools, and GAs in multi-objective optimization studies, where GAs effectively manages the trade-offs determined by PCA and Nftools. Through collaboration, the final designs are guaranteed to meet and surpass the standards established by the decision-makers, thereby improving energy efficiency, decreasing environmental impact, and extending equipment lifespan. A powerful and all-encompassing method was created when PCA, Nftool, and GAs were integrated to optimize the heat transfer equipment for a variety of industrial applications.

## 2. Materials and methods

### 2.1. Experimental setup and apparatus

**2.1.1. Shell and tube heat exchanger.** In Table 1, the notable features of an ultramodern shell-and-tube heat exchanger are presented along with the objectives of the research.

**2.1.2. Experimental setup.** Among the components of this heat exchanger are one shell, six tubes, and four stainless-steel baffles. Two motors were used in the heat exchanger for efficient water circulation. Fig 1 illustrates the experimental setup of the heat exchanger constructed with the specific characteristics listed in Table 1.

In this study, this investigated the use of hot and cold water to cool high-temperature streams. A Shell-and-Tube Heat Exchanger (STHE) consisting of a shell that circulated cooling water and a tube that circulated hot water was used to accomplish this. Laminar counterflow configurations exhibited superior efficiency over parallel flows when segmental baffles were used to enhance heat transfer. A diagram (Fig 1) shows the different orientations of the baffles inside the heat exchanger. Baffle spacing was found to play a crucial role in longitudinal flow efficiency, as excessive spacing resulted in less efficient flow. In addition, cross-flow and unsupported tube spans were cited as potential causes of tube failure owing to flow-induced vibrations.

Turbulent flow (Reynolds number > 1,000) caused a fluctuation in the shell heat transfer coefficients, with a power law equation with exponents ranging from 0.6 to 0.7. However, the

**Table 1. Heat exchanger setup specifications.**

| Parameters | Value |
| --- | --- |
| Physical shape parameter | |
| Heat transfer characteristics | Indirect contact |
| Heat exchanger span, L | 600 mm |
| Inner shell size, Di | 90 mm |
| Tube exterior diameter, Do | 20 mm |
| Quantity of tubing, Nt | 6 |
| Baffle population, Nb | 2 |
| Material class | SS METAL |

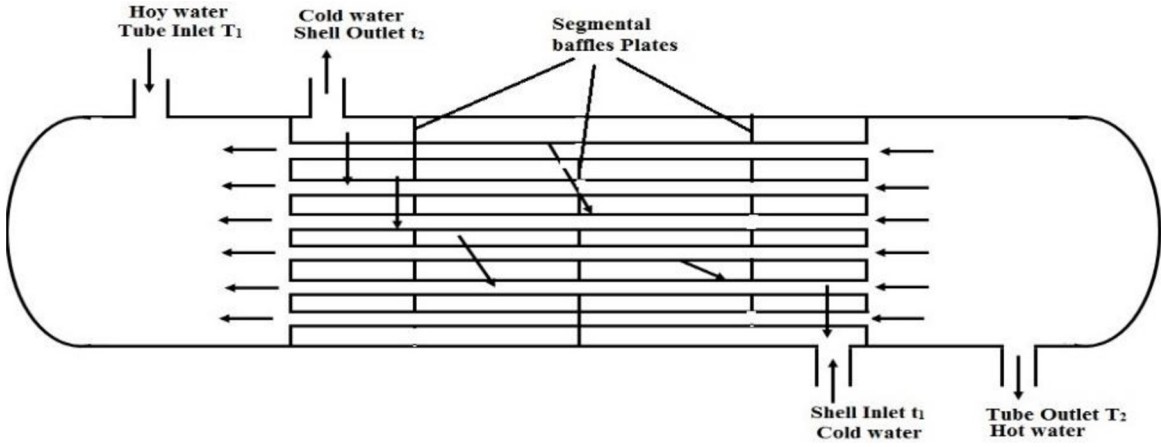

**Fig 1. An illustration of a shell and tube heat exchanger.**

pressure decrease showed a distinct power-law relationship with velocities greater than 1.7. The laminar flow produced a pressure decreases that were 0.33 and larger than the heat transfer coefficients. The heat transfer coefficients were greater than the pressure decreases with baffle spacing. It is recommended that the baffle spacing be set between 0.3 and 0.6 to maximize pressure drops, temperature differentials, and heat transfer efficiency.

**2.1.3. Uncertainty analysis.** Uncertainty is determined by combining the effects of individual inputs using the root sum square method, as defined by Kline and McClintock [48]. The measured data with quantifiable uncertainties included temperature readings of thermocouples, heat exchanger dimensions, and coil and shell side flow meter mass flow rates. The manufacturer provides an uncertainty of ±0.01 mm for the inside and outside tube diameters. In all experiments, the uncertainties were ±2.5% for the Reynolds number on the tube side, ±2.8% for the Nusselt number, 1.9% for the overall heat transfer coefficient, and ±2.2% for effectiveness.

## 2.2 Experimental parameters, design of experiments and methodology

**2.2.1 Input parameters.** This study examined the operation of a counterflow shell-and-tube heat exchanger by adjusting a number of crucial factors, such as the Cold-Water Flow Rate, Hot Water Temperature (T1), and Cold Water Inlet Temperature (t1).

The following input parameters were considered in the experimental design:

Cold Water Flow Rate (kg/min): 1, 2, 3, and 4

Hot Water Temperature (T1) in ˚C: 65, 70, 75, and 80

Inlet Temperature of Cold Water (t1) in ˚C: 30

**2.2.2 Design of Experiments (DOE).** Using the Design of Experiments (DOE) approach, Minitab software was used to systematically manipulate the model parameters. In an experiment using four factors, a 4-level Taguchi method experimental design was used [49]. The experiment consisted of sixteen runs. The results were replicated three times to ensure robustness and statistical validity.

**2.2.3 Test conditions.** In Table 2, the variables (flow rates) and their corresponding grades for the heat exchanger experiment are summarized. Each experiment was not explicitly

**Table 2. Test conditions of the heat exchanger experiment.**

| Attributes | Variables(kg/min) | Grades | | | |
|---|---|---|---|---|---|
| | | 1 | 2 | 3 | 4 |
| A | Flow rate of hot liquid | 1 | 2 | 3 | 4 |
| B | Flow rate of cold liquid | 1 | 2 | 3 | 4 |

described as to how it should be executed or how input parameters should be controlled. Table 2 summarizes the main factors that influence the output response. An experimental study of a heat exchanger is presented in Table 3.

In a counterflow shell-and-tube heat exchanger, High-Temperature Fluid (HTF) and Low-Temperature Fluid (LTF) exchange heat by convection. Each fluid had a mass flow rate ranging from 1 kg/min to 4 kg/min.

**2.2.4. Empirical investigation.** As shown in Table 3, the counterflow shell-and-tube heat exchanger was empirically examined in terms of runs, parameters, and results. The temperature readings in this study were made easier with the use of thermocouples, and cross-verification was conducted using digital thermometers. The flow rates of the hot and cold water within the heat exchanger system were measured using a digital flow meter.

**2.2.5. Methodology.** This approach defines goals and restrictions before methodically optimizing the issue. It may more efficiently explore the solution space and narrow down the input parameter space using Principal Component Analysis (PCA). By developing mathematical models of the underlying system behavior, RSM aids in the identification of ideal design areas. This tool uses artificial neural networks to capture nonlinear interactions between inputs and outputs to improve modeling accuracy. Subsequently, the possible solutions are iteratively refined using Genetic Algorithms (GA). We can incorporate the advantages of RSM, NFTOOL, and GA, comprehensively explore the parameter space, and guarantee robust convergence using a hybridized technique. This technique provides a deep knowledge of complicated systems, improves the overall optimization process, and produces dependable outcomes.

**Table 3. Empirical investigation of the heat exchanger.**

| S.No. | mh | mc | T1 | t1 | T2 | t2 | € | TR | U |
|---|---|---|---|---|---|---|---|---|---|
| | Kg/min | Kg/min | ˚C | ˚C | ˚C | ˚C | | ˚C\W | (W/m²ᵒC) |
| 1 | 1 | 1 | 80 | 30 | 51 | 58 | 0.57 | 0.76 | 4.91 |
| 2 | 1 | 2 | 75 | 30 | 53 | 51 | 0.72 | 0.63 | 5.11 |
| 3 | 1 | 3 | 70 | 30 | 57 | 47 | 0.70 | 0.76 | 4.15 |
| 4 | 1 | 4 | 65 | 30 | 54 | 44 | 0.83 | 0.58 | 4.77 |
| 5 | 2 | 1 | 75 | 30 | 54 | 49 | 0.66 | 0.78 | 4.36 |
| 6 | 2 | 2 | 80 | 30 | 52 | 56 | 0.54 | 0.85 | 8.69 |
| 7 | 2 | 3 | 65 | 30 | 44 | 47 | 0.69 | 0.61 | 11.26 |
| 8 | 2 | 4 | 70 | 30 | 42 | 52 | 0.97 | 0.28 | 19.48 |
| 9 | 3 | 1 | 70 | 30 | 50 | 49 | 0.96 | 0.37 | 6.94 |
| 10 | 3 | 2 | 65 | 30 | 55 | 52 | 0.61 | 0.79 | 8.66 |
| 11 | 3 | 3 | 80 | 30 | 56 | 57 | 0.51 | 0.96 | 11.57 |
| 12 | 3 | 4 | 75 | 30 | 51 | 48 | 0.56 | 0.92 | 11.61 |
| 13 | 4 | 1 | 65 | 30 | 52 | 42 | 0.87 | 0.52 | 5.01 |
| 14 | 4 | 2 | 70 | 30 | 53 | 55 | 0.84 | 0.44 | 13.23 |
| 15 | 4 | 3 | 75 | 30 | 54 | 52 | 0.56 | 0.91 | 11.88 |
| 16 | 4 | 4 | 80 | 30 | 52 | 57 | 0.55 | 0.82 | 18.10 |

Step-by-step process for using a combined RSM-Neural Fitting Tool (NFTOOL)-genetic algorithm (GA) approach for optimization:

**Step 1: Define the Problem**

- Clearly define the optimization problem, including objectives and constraints.

- Identify input parameters that can be adjusted to optimize the response or objective function.

**Step 2: Collect Data**

- Collect relevant data for the problem, including input parameters and responses.

- Ensure that the data are accurate and cover the range of parameter values.

**Step 3: Principal Component Analysis (PCA)**

- Retain the most important data while reducing the dimensionality of your data with PCA.

- The principal components of the data that capture variance should be identified and selected.

**Step 4: Response Surface Methodology (RSM)**

- Design and conduct experiments to collect data efficiently. It typically uses an experiment designed with different input parameter combinations.

- Fit an initial response surface model using statistical methods, such as regression analysis.

- Input parameters and response need to be evaluated to determine the accuracy of the response surface model's representation.

**Step 5: Neural Fitting Tool (NFTOOL)**

- Preprocess and normalize the collected data as required. Ensure that the data are in a format suitable for NFTOOL.

- Start MATLAB and open NFTOOL.

- Configure a neural network using NFTOOL. We define input variables, output variables, network architecture (e.g., number of hidden layers and neurons), and training options (e.g., training algorithm and number of epochs).

- Training a neural network using pre-processed data. The training process was monitored to ensure convergence.

**Step 6: Genetic Algorithms (GA)**

- Define an optimization problem. This involves specifying an objective function to maximize or minimize any constraints on the input parameters.

- Initialize the population of potential solutions (combinations of input parameters).

- Create probabilistic solutions using genetic operators such as selection, crossover, and mutation.

- Evaluate the fitness of each solution using a trained neural network model from NFTOOL. Fitness evaluation evaluates the response to each input parameter.

- Iteratively apply selection and genetic operators to develop a population of solutions. This process was repeated until the convergence or stopping criterion was satisfied.

**Step 7: Check Convergence**

- Determine whether GA has reached the optimal solution. Convergence may be defined as a change in the value of the objective function or by a certain number of generations.

**Step 8: Extract the Optimal Parameters**

- Retrieve the best combination of input parameters found by the GA. These parameters are the optimal solutions that maximize or minimize the objective function.

**Step 9: Assess Response**

- Use a response surface model created using RSM or a trained neural network model generated by NFTOOL to estimate the response associated with the correct input parameters.

**Step 9: Evaluate the practicality and Finish**

- Assess whether the optimized parameters are practical and feasible for real-world implementation.

- Process complete. The correct input parameters and expected responses were obtained based on the RSM-ANN-GA approach.

This program illustrates dynamic features based on statistics such as Principal Component Analysis (PCA) and Response Surface Methodology (RSM). PCA and RSM adapt to different inputs when modelling relationships within the data. By computing eigenvalues and eigenvectors, PCA algorithms can adapt to changes in data. NFtool, an easily accessible tool that allows users to create and optimize neural networks dynamically and intuitively, was integrated into MATLAB. The iterative improvement of solutions is another characteristic of a Genetic Algorithm (GA), which adds another layer of dynamics. By incorporating adaptive analyses, crossings, mutations, and replacements into the algorithm, this dynamic optimization process can keep up with changing conditions and continuously improve. In Fig 2 clearly represent the flow chart of the optimization.

## 2.3. Heat transfer analysis

**2.3.1. Overall heat transfer.** The log-mean temperature difference (LMTD) method was used to calculate the heat transfer rate (Q) in shell and tube heat exchangers [50].

$$Q = F \times U \times A \times \Delta T_{lm} \tag{1}$$

Where
Q is the rate at which heat is transferred (W or kW).
F is the Correction Factor
U is the overall heat transfer coefficient (W/m$^2$K)
A is the effective heat transfer area (m$^2$)
$\Delta T_{lm}$ is the log-mean temperature difference (k), calculated as:

$$\Delta T_{lm} = \frac{\Delta T_1 - \Delta T_2}{ln\left(\frac{\Delta T_1}{\Delta T_1}\right)} \tag{2}$$

Where as $\Delta T_1$ is the inlet of hot and cold fluid temperature difference at one end (k), $\Delta T_2$ is the outlet of hot and cold fluid temperature difference at one end (k).

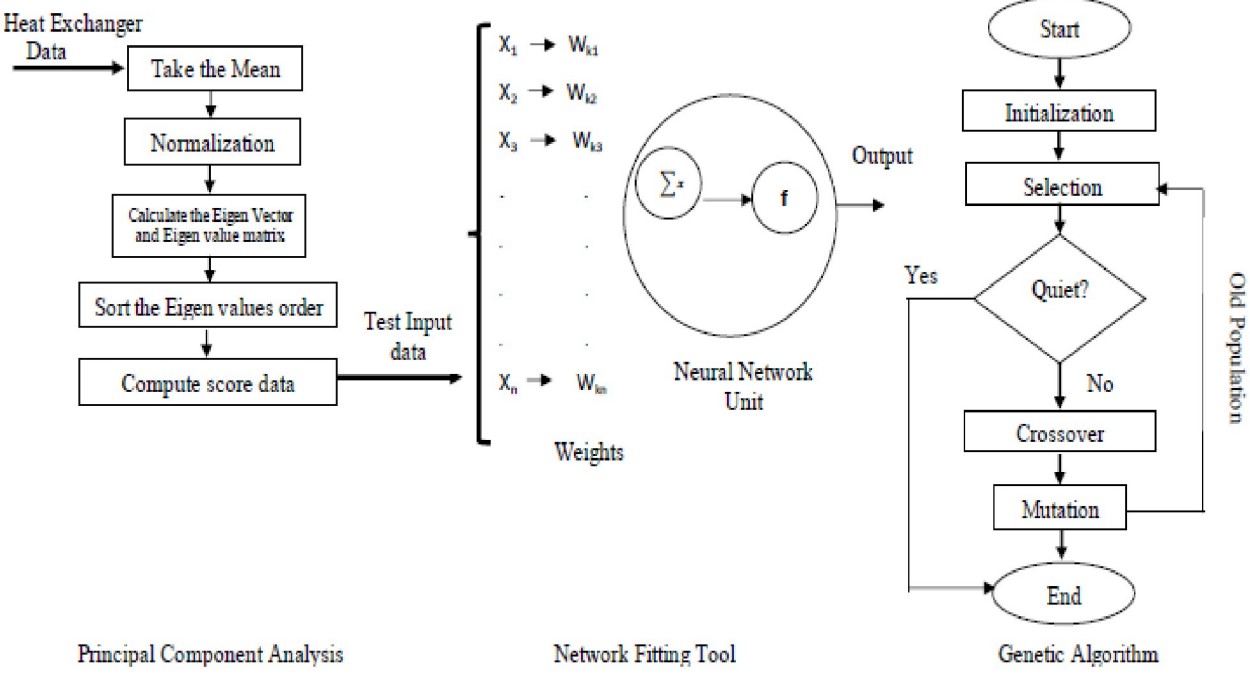

**Fig 2. Flow chart.**

**2.3.2. Thermal resistance and effectiveness.** Following are the formulas that can be used to calculate the values of effectiveness (ε) and thermal resistance (R) heat exchangers [51].

$$R = \frac{1}{UA} \tag{3}$$

$$\varepsilon = \frac{1 - (e^{-R})}{1 - (e^{-N*R})} \tag{4}$$

Where R is the thermal resistance (k/W)
U is the overall heat transfer coefficient (W/m2K)
A is the effective heat transfer area (m2)
N is the number of heat transfer units, calculates as:

$$N = \frac{UA}{C_{min}} \tag{5}$$

$C_{min}$ is the minimum value of mass flow rate of heat capacity of two fluids

## 2.4. Statistical analysis and modeling

Data are presented as mean values ± standard deviation. Principal component analysis (PCA) was performed using Minitab 18 to optimize grouping and provide an overview of the relationships between different extracts and their antioxidant activity. In addition, Minitab was used to generate 18 experimental cases and subsequent statistical analyses using response surface methodology (RSM). The Neural Network Toolbox™ in MATLAB 2015b was used for the optimization process.

**2.4.1. Principal Compound Analysis (PCA).** The relationships between different extracts and antioxidant activity were analyzed by Principal Component Analysis (PCA) using Minitab 18 [52–54]. An analysis of the principal components reduces data dimensionality and identifies key variance components. The following mathematical expressions represent PCA:

*2.4.1.1. Covariance Matrix (S).* The covariance matrix summarizes the variable relationships:

$$S = \frac{1}{n-1} \sum\nolimits_{i=1}^{n} (X_i - \breve{X})(X_i - \breve{X})^T \tag{6}$$

Where s is the representation of covariance matrix

n is the representation of number of data point

$X_i$ is the representation of data points

$\breve{X}$ is the representation of significance of mean of the data

*2.4.1.2. Eigen values and Eigen vectors.* Eigenvalues are a unique set of scalar values connected to a set of linear equations that are most likely to be seen in the matrix equations. The characteristic roots are another name for the eigenvectors. It is a nonzero vector that, following linear transformations, can only be altered by its scalar factor. PCA calculates the eigenvalues ($\lambda$) and eigenvectors ($v$) Covariance matrix:

$$\boldsymbol{S}\boldsymbol{v} = \lambda\,\boldsymbol{v} \tag{7}$$

**2.4.2. Response Surface Methodology (RSM).** The response of the system to the parameters was modeled using Response Surface Methodology (RSM). RSM is modeled using a modified cubic polynomial, as mentioned in the following section:

$$Y = \boldsymbol{\beta}_0 + \sum\nolimits_{i=1}^{3} \boldsymbol{\beta}_i X_i + \sum\nolimits_{i=1}^{3} \boldsymbol{\beta}_{ii} X_i^2 + \sum\nolimits_{i<j}^{3} \boldsymbol{\beta}_{ij} X_i X_j + \sum\nolimits_{i<1}^{3} \boldsymbol{\beta}_{iij} X_i^2 X_j \tag{8}$$

Where Y is representation of the response variable

$X_i$ is representation of the independent variable

$\beta_0$, $\beta_i$, $\beta_{ii}$, $\beta_{ij}$, $\beta_{iij}$ are representation of the coefficients

**2.4.3. Neural Net Fitting (NFtool).** A graphical user interface referred to as 'nftool' is included in MATLAB's Neural Network Toolbox for creating, training, and evaluating neural networks [55]. It simplifies the process of defining network architectures, preparing datasets, selecting algorithms, and assessing model performance using FacenfTool. The model performance was evaluated using the Mean Squared Error (MSE) and other relevant metrics. Users without coding experience can easily develop neural networks using this user-friendly tool."

## 2.5. Optimization using Nftool and Genetic Algorithm (GA)

Multiple fields have used GA to optimize the algorithms. An Nftool-GA hybrid can be used in ultrasonic extraction procedures, as reported previously. ANNs were constructed first, followed by GAs, crossovers, and mutations, until optimal results were obtained [56].

Genetic algorithms were used to optimize the system. The following steps are included in a typical GA algorithm:

1. Initialization: Chromosomes were generated as candidates in the initial population.

2. Selection: Chromosomes were evaluated for fitness and parents were selected for reproduction based on fitness.

3. Crossover (recombination): Genetic information from selected parents is combined to create a new offspring (chromosomes).

4. Mutation: Small random changes are introduced into offspring chromosomes.

5. Replacement: The offspring of the old population are replaced by the new population.

6. Conclusion: Steps 2–5 are repeated for predefined generations or until convergence.

The specific formulation and parameters of the GA are tailored to the optimization problem at hand.

## 3. Result and discussion

### 3.1. Principal component analysis

To uncover latent patterns and relationships within the heat exchanger dataset, Principal Component Analysis was applied to parameters including mass flow rates (mh and mc), temperatures (T1, t1, t2, and t2), and additional variables (TR). The purpose of this analytical approach is to reduce dimensionality and identify the relevant components that explain variance in the data.

**3.1.1. Eigenvalues.** Eigenvalues that represent the variance explained by each principal component were primary indicators. Components with eigenvalues greater than one were retained, adhering to the Kaiser criterion. For the first nine components of this dataset, the eigenvalues were as follows:

An analysis of principal components was performed on the shell and tube heat exchangers, as shown in Fig 3 based on the Table 4. There is an eigenvalue greater than 1 for each of the first four principal components in these results. These four components accounted for 87.7% of data variation. For the fourth principal component, the eigenvalues formed a straight line. There is an 87.7% explanation of variance in the data based on the first three principal components.

**3.1.2. Proportion of variance.** It is important to understand how each principal component contributes to the overall variance of the data. A notable explanatory capacity is reflected in these proportions. According to this dataset, the proportions were as follows:

PC1: 35.1%

PC2: 26.1%

PC3: 14.4%

PC4: 12.1%

PC5: 5.7%

PC6: 4.1%

PC7: 2.2%

PC8: 0.2%

PC9: 0.1%

To understand the overarching variance, it is helpful to determine the weight of each component.

**3.1.3. Cumulative variance.** To determine the number of components to be retained, the cumulative proportion of variance is estimated using successive principal components. These were the cumulative proportions used in this analysis.

• PC1–PC9 cumulatively accounted for 100% of the variance in the dataset.

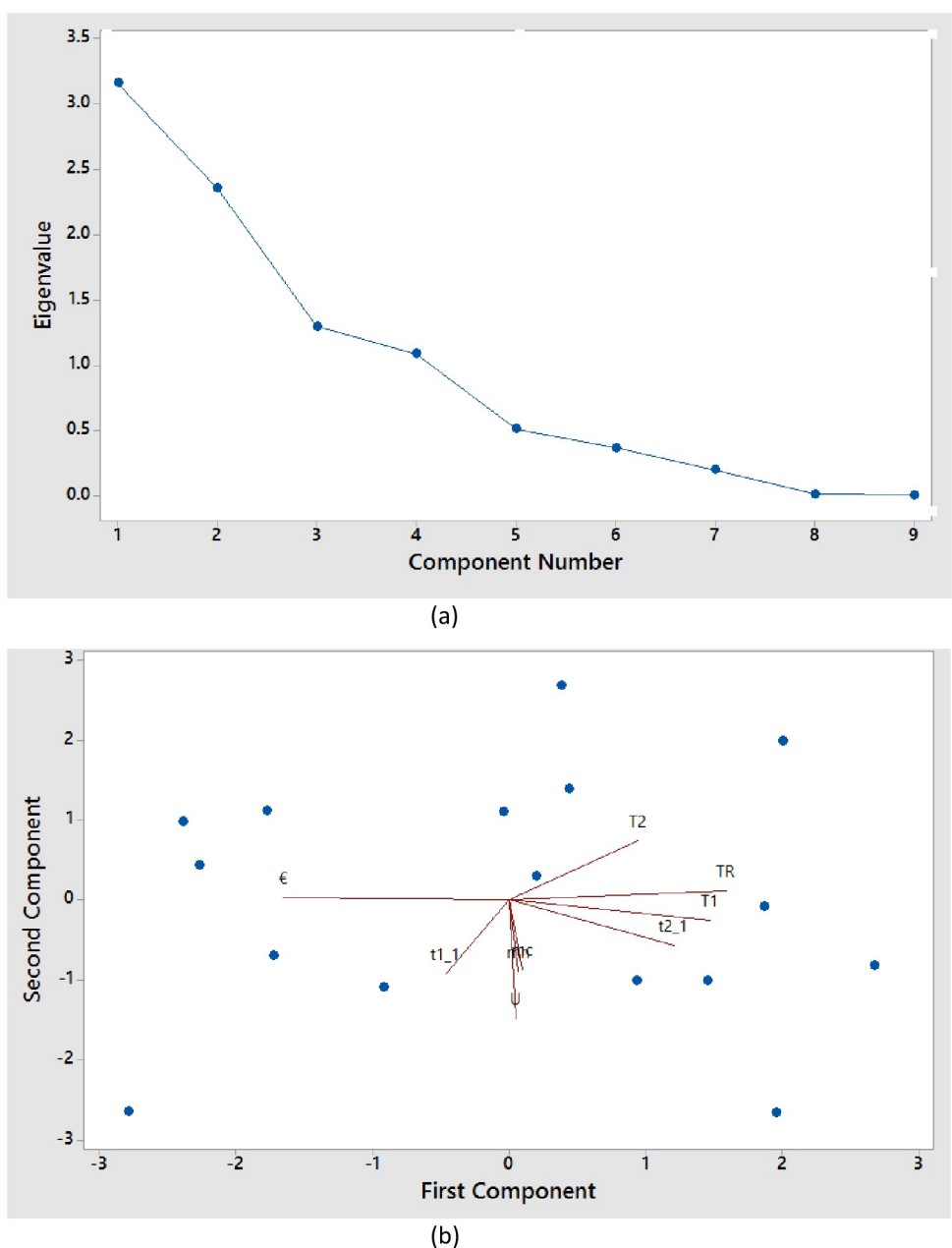

**Fig 3. Observation plot (A) and biplot (B) of the first two components produced by principal component analysis (PCA) of the shell and tube heat exchangers.**

**Table 4. Eigen analysis of the correlation matrix.**

| Eigenvalue | 3.158 | 2.352 | 1.298 | 1.087 | 0.511 | 0.368 | 0.201 | 0.016 | 0.006 |
|---|---|---|---|---|---|---|---|---|---|
| Proportion | 0.351 | 0.261 | 0.144 | 0.121 | 0.057 | 0.041 | 0.022 | 0.002 | 0.001 |
| Cumulative | 0.351 | 0.612 | 0.757 | 0.877 | 0.934 | 0.975 | 0.997 | 0.999 | 1.000 |

**Table 5. Eigenvectors.**

| Variable | PC1 | PC2 | PC3 | PC4 | PC5 | PC6 | PC7 | PC8 | PC9 |
|---|---|---|---|---|---|---|---|---|---|
| mh | 0.021 | -0.382 | -0.577 | -0.279 | 0.418 | -0.317 | -0.236 | 0.331 | -0.051 |
| mc | 0.030 | -0.369 | 0.173 | 0.734 | 0.202 | 0.268 | -0.108 | 0.408 | -0.059 |
| T1 | 0.465 | -0.106 | 0.223 | -0.257 | -0.207 | 0.240 | -0.738 | 0.000 | 0.103 |
| t1_1 | -0.147 | -0.388 | -0.476 | -0.015 | -0.669 | 0.368 | 0.132 | 0.011 | -0.003 |
| T2 | 0.298 | 0.318 | -0.447 | 0.076 | 0.404 | 0.593 | 0.085 | -0.222 | 0.182 |
| t2_1 | 0.381 | -0.238 | 0.294 | -0.436 | 0.108 | 0.258 | 0.532 | 0.296 | -0.269 |
| € | -0.523 | 0.017 | 0.079 | -0.184 | 0.205 | 0.372 | -0.270 | -0.157 | -0.638 |
| TR | 0.503 | 0.051 | -0.213 | 0.295 | -0.149 | -0.276 | 0.002 | -0.271 | -0.664 |
| U | 0.015 | -0.629 | 0.152 | 0.014 | 0.234 | -0.044 | 0.085 | -0.698 | 0.171 |

This indicates that the total variation in the dataset was effectively captured by all nine principal components.

**3.1.4. Eigenvectors.** The coefficients of the eigenvectors determine each principal component's linear combination of the variables. Based on these coefficients, we were able to understand the magnitude and direction of the variables' effects on each component. The interpretation of these coefficients in the context of the heat-exchanger parameters was simplified by examining these coefficients. In Table 5, denotes the eigenvector.

## 3.2. Response surface methodology modeling

The analysis consisted of three different response surface regression investigations conducted using the Minitab software framework. Each analysis focused on a particular response variable (TR, U) while considering the predictor variables (mh, mc, T1, t1_1, T2, t2_1). The key findings of each analysis are summarized as follows:

In the context of the analysis conducted, focusing on the relationship between €, TR & U and the predictor variables (mh, mc, T1, t1_1, T2, t2_1), the following results and outcomes are reported:

**3.2.1. Response surface regression: €.**

- R-squared (R-sq):99.53% indicates that the model accounted for approximately 99.53% of the variance in.

- Taking model predictors into account, adjusted R-squared is 98.25%.

- R-squared for prediction (R-sq(pred)): denoted by * indicates strong predictive capabilities of the model.

- The model exhibited significance (p-value = 0.000), indicating its ability to explain a substantial portion of the variance in €.

- Regression equation for effectiveness is:

$$€ = 0.7769 - 0.07604 \text{ mh} - 0.3846 \text{ T1} - 0.931 \text{ T2} + 0.3407 \text{ t2\_1} -$$
$$0.0488 \text{ mh*mh} + 0.1411 \text{ mc} * \text{mc} - 0.4855 \text{ T1} * \text{T1} - 0.0878 \text{ T2} * \text{T2} -$$
$$0.0668 \text{ mh} * \text{T1} + 0.4621 \text{ T1} * \text{t2\_1} + 0.926 \text{ t1\_1} * \text{T2}$$

### 3.2.2. Response surface regression: TR.

- R-squared (R-sq):69.97% indicates that the model explains approximately 69.97% of the variance in TR.

- Model adjusted R-squared (R-sq(adj)):62.47—Taking into account predictors of the model, adjusted R-squared value is 62.47%.

- R-squared for prediction (R-sq(pred)):51.64% demonstrates the predictive capacity of the model.

- This model contributed significantly to TR variance as indicated by the p-value of 0.002 in the ANOVA table.

- Regression equation for thermal resistance is

$$TR = 0.5590 + 0.1165 * T1*T1 + 0.2099 * mh * mc + 0.2144 * t1\_1 * T2$$

### 3.2.3. Response surface regression: U.

- R-squared (R-sq): 99.81%—reflects the model's ability to explain approximately 99.81% of the variance in U

- In order to account for model predictors, R-squared (R-sq(adj)) has been adjusted to 99.59%.

- R-squared for prediction (R-sq(pred)): 98.25%—demonstrates the predictive prowess of the model.

- With a p-value of 0.000, the analysis of variance table indicates that the model is highly significant, explaining a considerable part of the variance.

- Regression equation for overall heat transfer coefficient is

$$U = -2.6 + 9.31 * mh + 15.30 * mc - 1.221 * T2 + 0.2550 * t2\_1 +$$
$$0.01926 * T2 * T2 - 0.2091 * mh * T2 + 0.0756 * mh * t2 * 32 * T2\_60$$

The €, TR and U models demonstrate significance, explaining a substantial portion of the variance in their respective response variables. The main terms contributing to the models included linear, squared, and 2-way interactions. The research effort began with the use of Minitab software to conduct a rigorous and comprehensive response surface regression (RSR) analysis. This statistical exploration aims to uncover complex relationships between multiple predictor variables, including mh, mc, T1, t1_1, T2, and t2_1, and the response variable €, TR and U. Using precise stepwise term selection, the research team constructed a parsimonious model adhering to predefined significance levels to distill the essential components of this complex relationship.

For the purpose of determining the validity of the results, data were analyzed using a comprehensive analysis of variance approach. Model summaries, complete with R-squared values and adjusted R-squared values, provide a clear overview of the explanatory power and predictive accuracy of the model. Coded coefficients and the regression equation, expressed in coded units, facilitate quantitative interpretation and estimation. As shown in Table 3, the experimental results are consistent with the RSM predictions.

This research approach goes beyond traditional statistics by using graphical tools such as the simple probability plot, the fit plot, histograms, and order plots, as shown in Fig 4. This

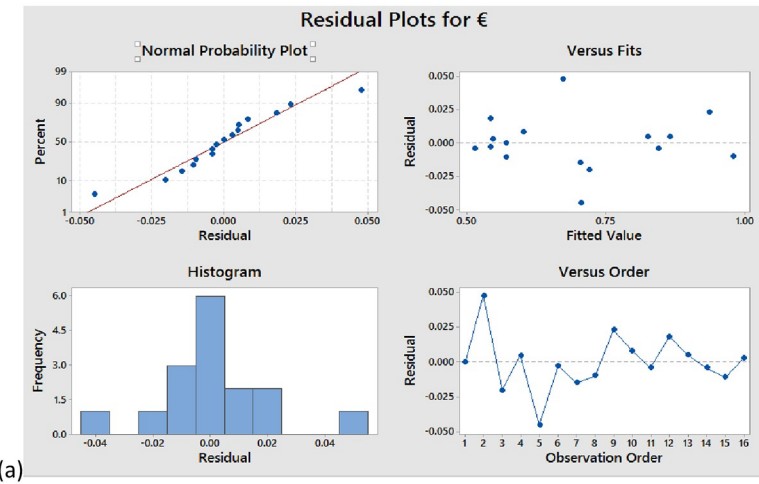

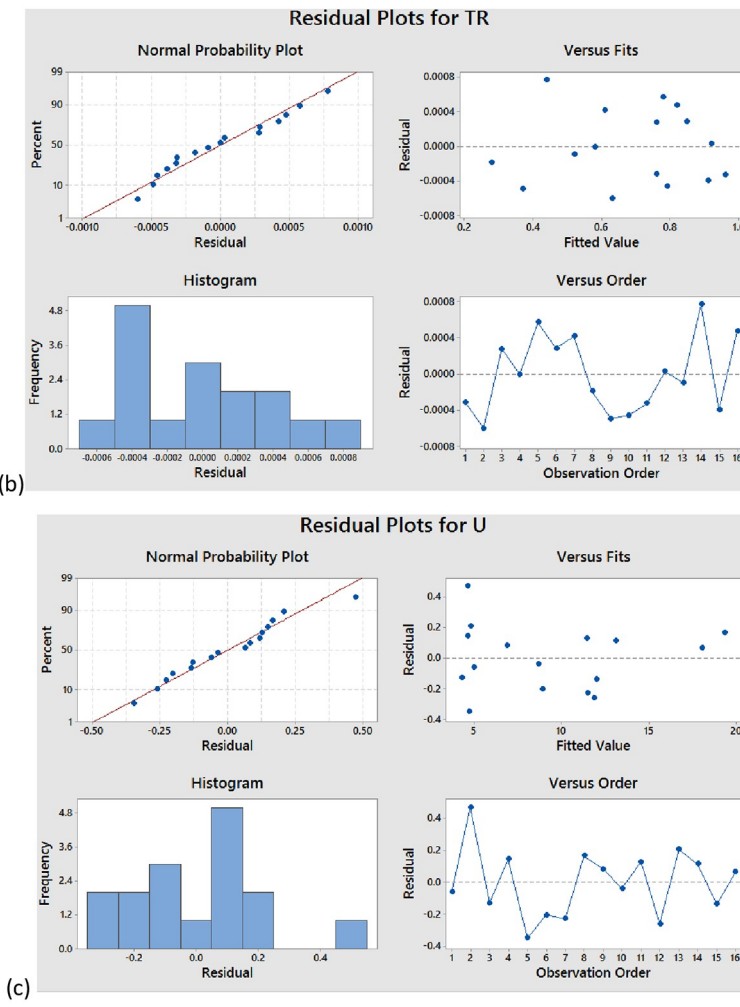

**Fig 4. Represents the Normal probability plot, histogram, residuals versus fits and residuals versus order for (a) effectiveness (b) thermal resistance and (c) overall heat transfer coefficient.**

visual aid also aids in ensuring the results are robust and reliable by facilitating the evaluation of data normality.

### 3.3. Neural Network Fitting Tool (Nftool)

A neural network can be easily designed and analyzed using a Neural Fitting Tool (NFTOOL) in MATLAB. In the Nftool the division of the 16 samples into training (60%), validation (20%), and testing (20%) sets for heat exchanger applications. By using Bayesian regularization during training, NFTOOL prevents overfitting and enhances the generalization. As shown in Fig 6, the tool had an architecture. Figs 5 and 6 illustrate how this process facilitates the development and validation of neural networks for predictive tasks. the result of predict as shown in Table 6.

An effective neural network must be evaluated using a regression analysis, as illustrated in Fig 6. Predicted and actual outcomes were compared to evaluate the model's accuracy. An estimation of the network's accuracy and consistency was based on a regression plot of the data. Furthermore, these analyses highlighted discrepancies in the model, improving its reliability, performance, and effectiveness. It is noteworthy that the R values are highly indicative of

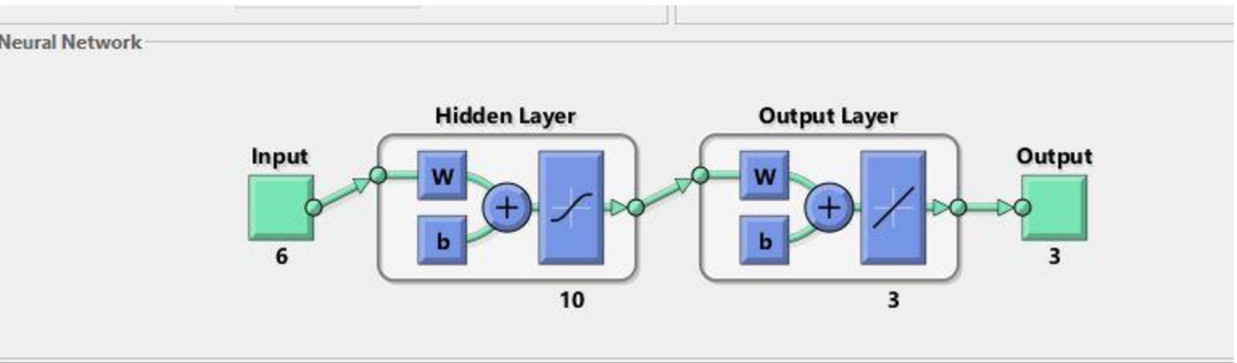

**Fig 5. Neural network architecture.**

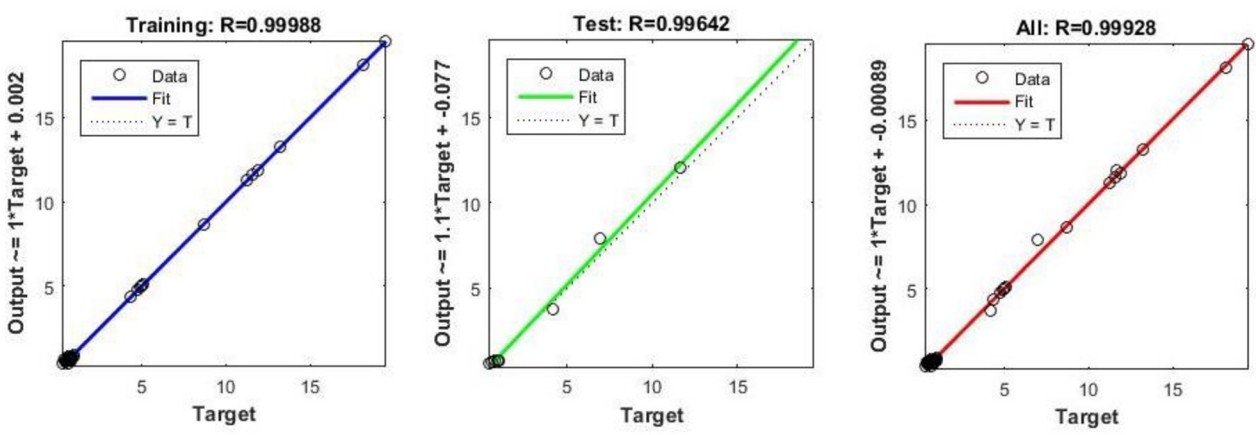

**Fig 6. Experimental and predicted regression of values.**

**Table 6. Matrix summary of investigated responses.**

| | Variables | | | | | | Responses | | | | | | | | |
|---|---|---|---|---|---|---|---|---|---|---|---|---|---|---|---|
| | | | | | | | | | Predict | | | Predict | | | Predict |
| S.No. | mh | mc | T1 | t1 | T2 | t2 | €  | | | TR | | | U | | |
| | Kg/min | Kg/min | ˚C | ˚C | ˚C | ˚C | | | | oC\W | | | W/m2oC | | |
| | | | | | | | Experimental | RSM | ANN | Experimental | RSM | ANN | Experimental | RSM | ANN |
| 1 | 1 | 1 | 80 | 30 | 51 | 58 | 0.57 | 0.57 | 0.596 | 0.76 | 0.843 | 0.843 | 4.91 | 4.971 | 4.923 |
| 2 | 1 | 2 | 75 | 30 | 53 | 51 | 0.72 | 0.7 | 0.657 | 0.63 | 0.742 | 0.742 | 5.11 | 4.638 | 5.11 |
| 3 | 1 | 3 | 70 | 30 | 57 | 47 | 0.7 | 0.708 | 0.7 | 0.76 | 0.716 | 0.716 | 4.15 | 4.276 | 3.708 |
| 4 | 1 | 4 | 65 | 30 | 54 | 44 | 0.83 | 0.834 | 0.784 | 0.58 | 0.594 | 0.594 | 4.77 | 4.623 | 4.772 |
| 5 | 2 | 1 | 75 | 30 | 54 | 49 | 0.66 | 0.666 | 0.658 | 0.78 | 0.771 | 0.771 | 4.36 | 4.706 | 4.352 |
| 6 | 2 | 2 | 80 | 30 | 52 | 56 | 0.54 | 0.555 | 0.577 | 0.85 | 0.77 | 0.77 | 8.69 | 8.893 | 8.682 |
| 7 | 2 | 3 | 65 | 30 | 44 | 47 | 0.69 | 0.692 | 0.834 | 0.61 | 0.495 | 0.495 | 11.26 | 11.487 | 11.264 |
| 8 | 2 | 4 | 70 | 30 | 42 | 52 | 0.97 | 0.969 | 0.828 | 0.28 | 0.288 | 0.288 | 19.48 | 19.314 | 19.474 |
| 9 | 3 | 1 | 70 | 30 | 50 | 49 | 0.96 | 0.945 | 0.755 | 0.37 | 0.516 | 0.516 | 6.94 | 6.857 | 7.921 |
| 10 | 3 | 2 | 65 | 30 | 55 | 52 | 0.61 | 0.602 | 0.754 | 0.79 | 0.809 | 0.809 | 8.66 | 8.696 | 8.657 |
| 11 | 3 | 3 | 80 | 30 | 56 | 57 | 0.51 | 0.501 | 0.541 | 0.96 | 0.885 | 0.885 | 11.57 | 11.441 | 11.571 |
| 12 | 3 | 4 | 75 | 30 | 51 | 48 | 0.56 | 0.569 | 0.678 | 0.92 | 0.685 | 0.685 | 11.61 | 11.868 | 12.028 |
| 13 | 4 | 1 | 65 | 30 | 52 | 42 | 0.87 | 0.869 | 0.794 | 0.52 | 0.537 | 0.537 | 5.01 | 4.801 | 5.012 |
| 14 | 4 | 2 | 70 | 30 | 53 | 55 | 0.84 | 0.859 | 0.707 | 0.44 | 0.602 | 0.602 | 13.23 | 13.113 | 13.228 |
| 15 | 4 | 3 | 75 | 30 | 54 | 52 | 0.56 | 0.552 | 0.629 | 0.91 | 0.771 | 0.771 | 11.88 | 12.015 | 11.886 |
| 16 | 4 | 4 | 80 | 30 | 52 | 57 | 0.55 | 0.547 | 0.592 | 0.82 | 0.957 | 0.957 | 18.1 | 18.034 | 18.099 |

model performance. For target R is 0.99988, test R is 0.99642, and total R is 0.99928, the experimental and predicted regression values when trained using NFTOOL are depicted in Fig 6.

Fig 7 shows the crucial training box in the Neural Fitting Tool (NFTOOL), which offers vital insights into model correctness and performance. Post-training efficacy was assessed using performance metrics, regression outcomes, and error histograms. Calculate prediction quality using performance metrics, Mean Squared Error (MSE), Root Mean Squared Error (RMSE), Mean Absolute Error (MAE) and Coefficient of Determination ($R^2$), predicted outputs ($\hat{y}$) and target outputs ($y$). MSE is expressed as:

$$MSE = \frac{1}{n} \sum (\hat{y} - y)^2 \tag{9}$$

The forecasted and desired outcomes were more closely aligned, with smaller MSE values. Regression coefficients express the associations between the expected ($\hat{y}$) and desired ($y$) outputs in the following equation:

$$\hat{y} = s_0 + s_1 x_1 + s_2 x_2 + \cdots + s_n x_n \tag{10}$$

where $\hat{y}$ denotes the forecasted output, b0 represents the intercept, and b1, b2,..., bn symbolize the regression coefficients, while x1, x2,..., xn signify the input variables.

Histograms visualizing the error distributions (e) between the predicted and target outputs using equation is

$$e = \hat{y} - y \tag{11}$$

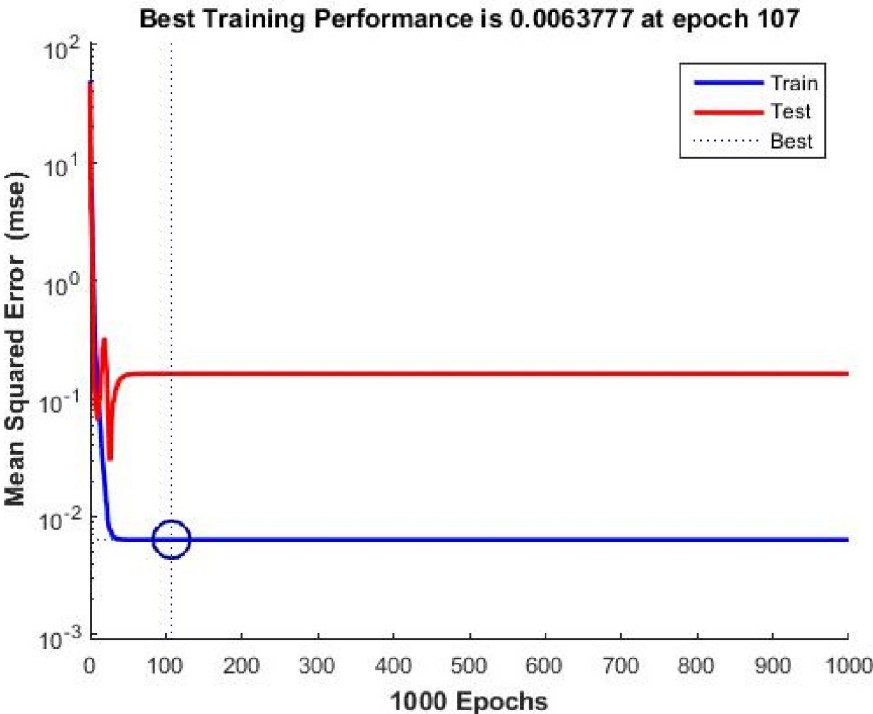

**Fig 7. Nftool best training performance.**

It is essential to evaluate and refine neural network models using performance plots such as those shown in Fig 7. Model performance can be improved through these insights, allowing for better predictions.

Fig 8 demonstrates the importance of tracking the progress of the network during training. Epoch counts, training errors, and validation errors are among the vital data conveyed by these methods. Analyzing the convergence and network performance enhances the overall progress by optimizing the training algorithms. Through a detailed examination of the training state, neural network models for diverse applications that are more accurate and effective can be developed.

The error histogram in Fig 9 shows the distributions of the predicted and actual values. Thus, bias or skewness can be detected, enabling the identification of patterns and outliers. Analyzing the model in this way enhances its predictive capability, reliability, and effectiveness.

With NFTOOL, the trained networks can be easily evaluated. The MSE, MAE, and RMSE metrics obtained post-training were used to assess the performance of the network and to measure its accuracy and prediction errors. Based on the data in Fig 10, it appears that the regression model is accurate in predicting the target data. According to Fig 11, the trained network generates a histogram of prediction errors. Analyzing the magnitude and frequency of errors provides insight into the predictive performance and biases or skew of the network. Training and testing were used in the development of the NFTool, resulting in predictions that were consistent with the measurements obtained for each response.

### 3.4. Genetic algorithms

This study uses multi-objective genetic algorithms (GA) to address a complex decision-making problem involving cost (€), thermal resistance (TR), and ultimate utility (U). By efficiently

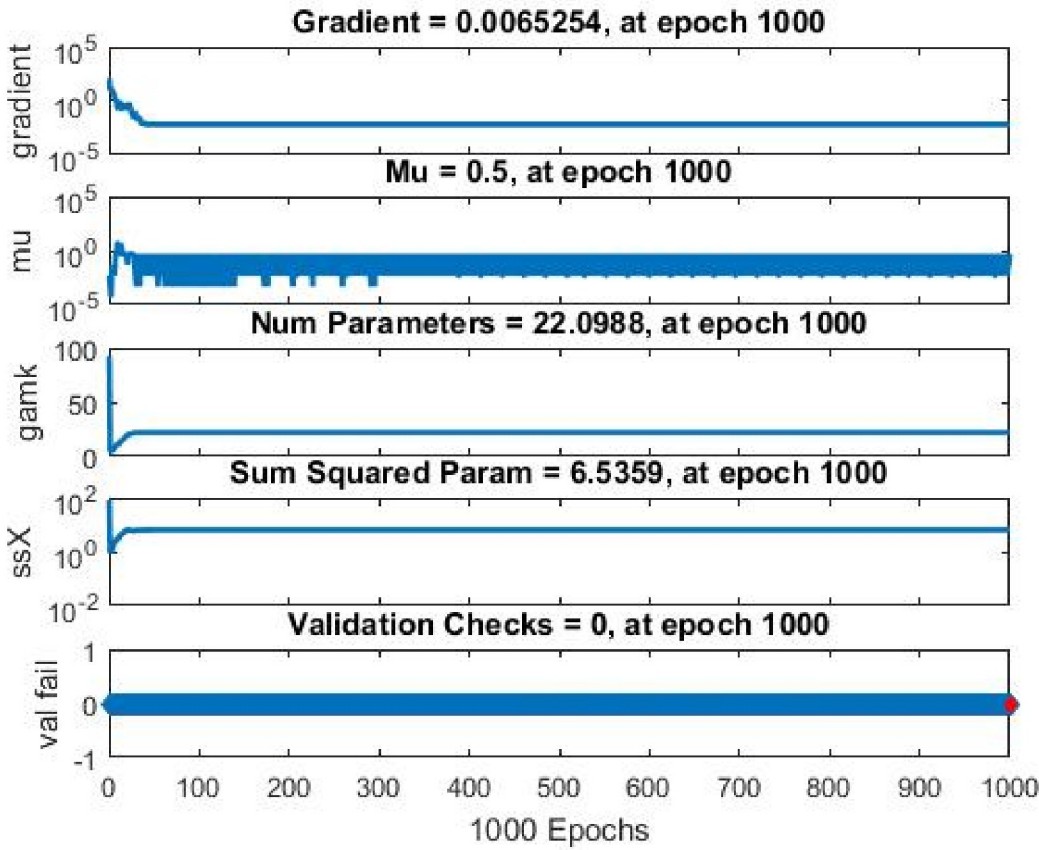

**Fig 8. NFTool's training state.**

exploring the solution space, the GA revealed a range of paraeto front solutions. The trade-offs between cost, thermal resistance, and overall utility emphasize the importance of making decisions in a balanced manner. As a result of the Pareto front analysis, optimal parameter combinations were identified, which provided valuable guidance for decision makers, as shown in Table 7.

Multi-objective GAs have demonstrated their effectiveness by efficiently converging on Pareto-optimal solutions while retaining computational efficiency. Hybrid GAs play a crucial role in optimizing input parameters. Fig 12 visually represents the Pareto front solutions, showing a delicate balance between the effectiveness, thermal resistance, and overall heat-transfer coefficient. Fig 13 shows the iteration versus fitness function.

Overall, the application of multi-objective GA has proven to be highly effective in solving complex engineering design problems with competing objectives. Pareto front solutions provide decision-makers with a range of versatile options to select design parameters that align with their unique needs and preferences, significantly contributing to the optimization and decision support field.

## 4. Conclusion

This study represents a paradigm shift in the optimization of shell and heat exchanger systems because it highlights the need for thermal engineering. This study aims to construct a novel

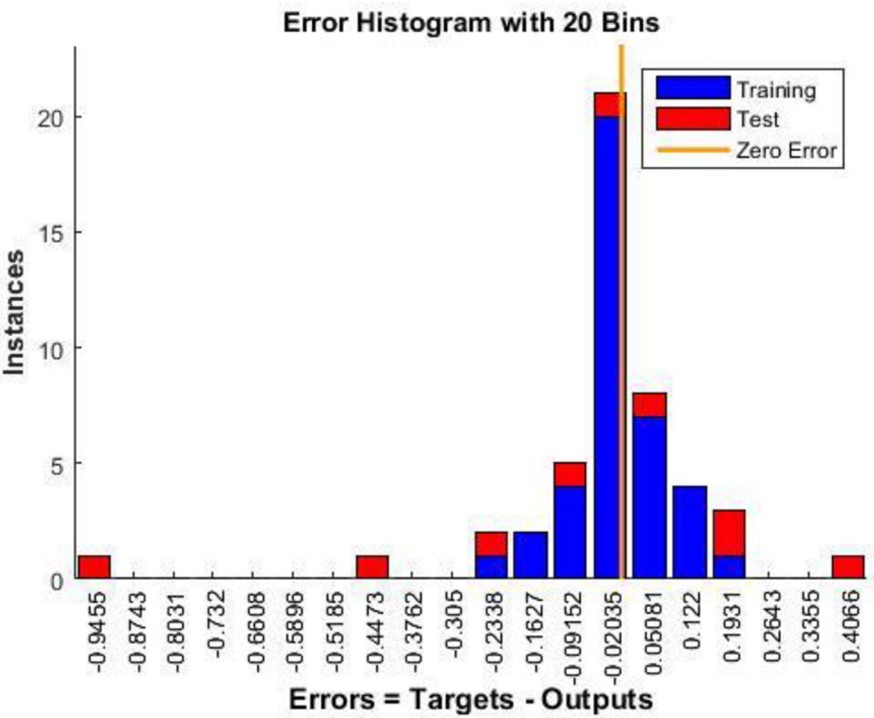

**Fig 9. Nftool error histogram.**

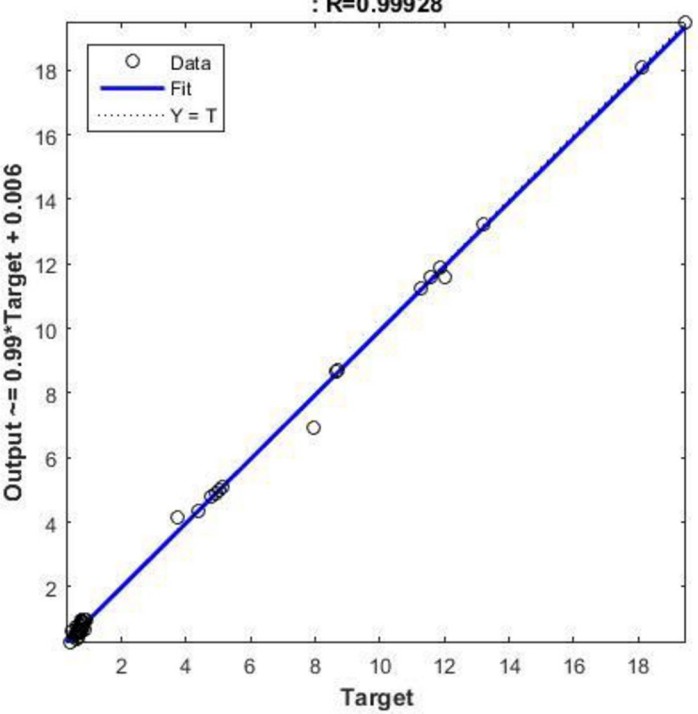

**Fig 10. Evaluate regression of values.**

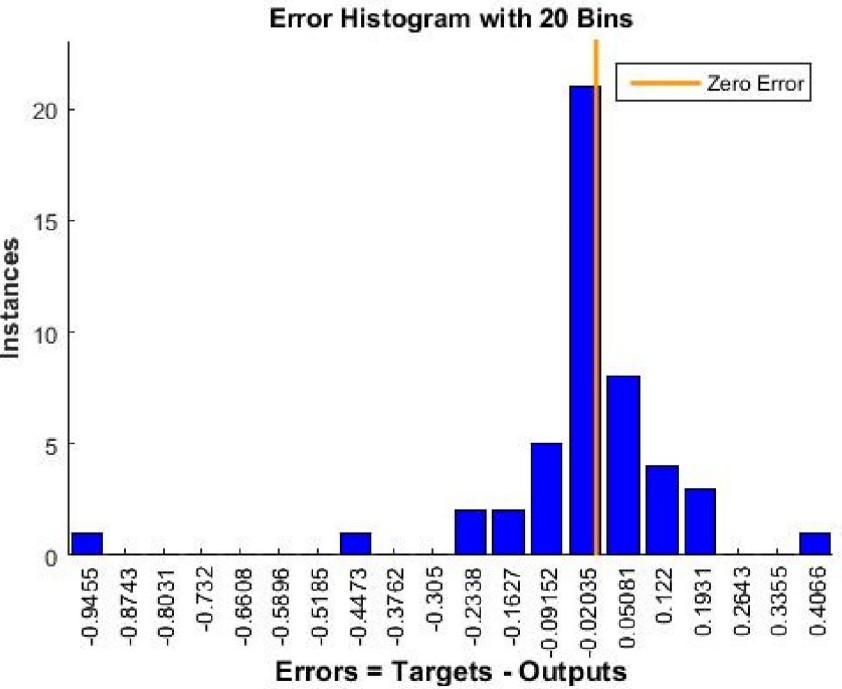

**Fig 11. Nftool evaluate error histogram.**

hybrid framework that includes cutting-edge approaches, such as Design of Experiments (DOE), Principal Component Analysis (PCA), Response Surface Regression (RSM), Neural Network Fitting Tools (NFTOOL), and Genetic Algorithms (GA). Conventional optimization techniques lack the depth and complexity that this method offers, which significantly increases efficiency.

The Neural Network Fitting Tool (NFTOOL) demonstrates a sophisticated design in terms of its predictive and accurate performance. It can be seen from the R-values, which are 0.99988 for the goal, 0.99642 for the test, and 0.99928 for the overall model, that the model adequately captures intricate interactions. As NFTOOL estimates the results almost exactly, it shows its ability to decode complex patterns in the data.

Genetic Algorithms (GA) are effective in exploring the solution space and uncovering the most economical, cost-effective, and heat-resistant solutions. The heat transfers for mh, mc, and T1 can be simplified by considering the physical mechanisms involved. By integrating PCA in accordance with the intricacy of input parameters, our hybrid architecture ensures a customized and effective optimization process. Further support for the stability of our methodology can be found in the best results obtained from the GA, NFTOOL, and PCA forecasts. In addition to pioneering new techniques for forecasting intricate correlations, NFTOOL provided Pareto-front solutions, whereas GA sought out hidden patterns that had previously been unknown, and PCA excelled at uncovering such patterns.

**Table 7. Optimal parameters input parameters by nftool-GA.**

| mh | mc | T1 | t1 | T2 | t2 | € | TR | U |
|---|---|---|---|---|---|---|---|---|
| Kg/min | Kg/min | ˚C | ˚C | (˚C) | (˚C) | | | |
| 1 | 1 | 79 | 30 | 52 | 56 | 0.59 | 0.68 | 5.08 |

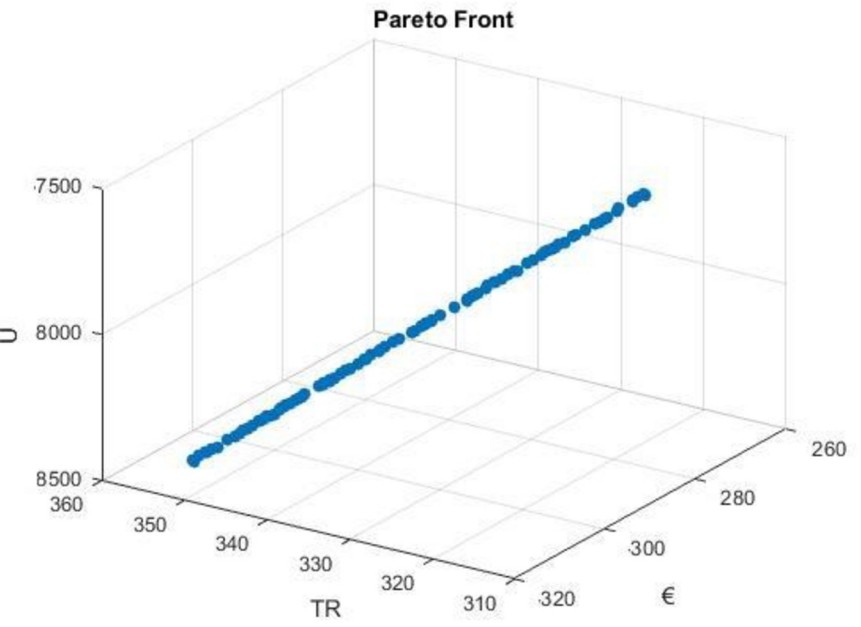

**Fig 12. Visually represents the Pareto front solutions.**

Through the combination of PCA, NFTOOL, and GA, this hybrid optimization methodology breaks through traditional barriers to usher in a new era in thermal engineering. It provides highly accurate, flexible, and refined forecasts using the PCA, NFTOOL, and GA techniques. Researchers, engineers, and industry practitioners will benefit greatly from these groundbreaking developments in heat-exchanger design.

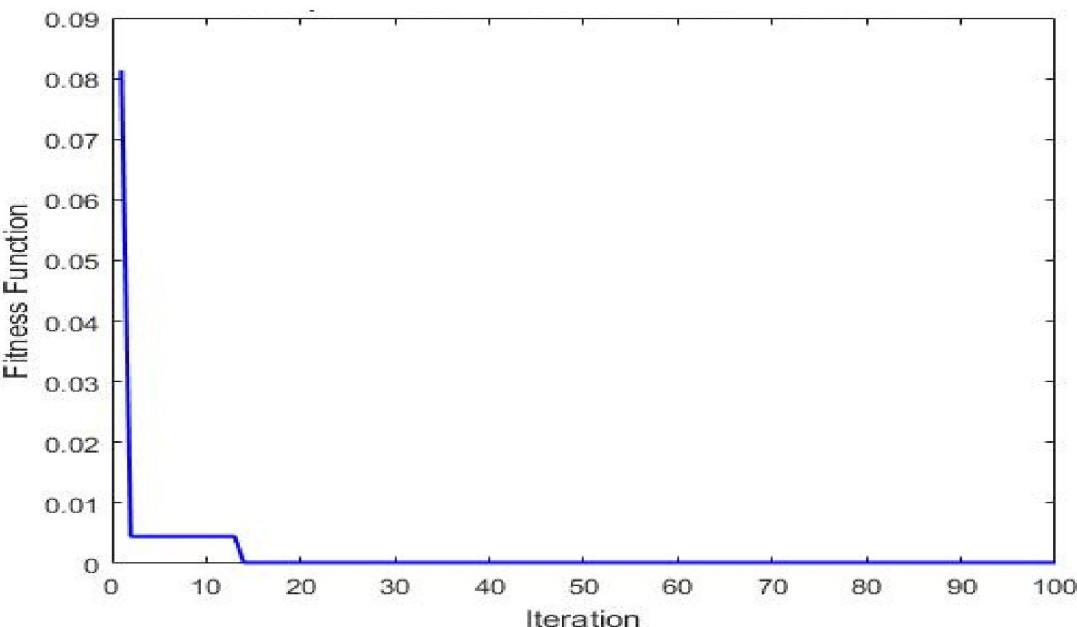

**Fig 13. Iteration versus fitness function.**

## Author Contributions

**Conceptualization:** Ajmeera Kiran, Mudassir Khan, Mohammad Khalid Imam Rahmani.

**Data curation:** Mudassir Khan, Laxmi Upadhyay.

**Formal analysis:** B. Venkatesh, Laxmi Upadhyay, T. Lakshmi Narayana.

**Funding acquisition:** Laxmi Upadhyay.

**Investigation:** B. Venkatesh, Laxmi Upadhyay.

**Methodology:** B. Venkatesh, T. Lakshmi Narayana.

**Project administration:** Mudassir Khan.

**Resources:** T. Lakshmi Narayana.

**Software:** J. Chinna Babu.

**Supervision:** Mudassir Khan.

**Validation:** Ajmeera Kiran, J. Chinna Babu.

**Visualization:** Ajmeera Kiran.

**Writing – original draft:** Mohammad Khalid Imam Rahmani.

**Writing – review & editing:** Mohammad Khalid Imam Rahmani, J. Chinna Babu.

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
