## [Decision Letter · Decision Letter 0]

24 Mar 2024

PONE-D-24-06376Performance optimization for an optimal operating condition for a Shell and Heat Exchanger Using a Multi-Objective Genetic Algorithm ApproachPLOS ONE

Dear Dr. Khan,

Thank you for submitting your manuscript to PLOS ONE. After careful consideration, we feel that it has merit but does not fully meet PLOS ONE’s publication criteria as it currently stands. Therefore, we invite you to submit a revised version of the manuscript that addresses the points raised during the review process.

**The author should critically observed the comments provided by the esteem reviewers and provide the response to their comments. The following are some major comments which need to focus will addressing the issues raise by the reviewers such as****1. Elaborate the heat loss in experimental setup.****2. Add the Error analysis of the experimental setup measuring instruments.****3. Huge gap between the present work and the conclusion.****4. Why the optimization technique  (specific type) has been selected for the present work.**

We look forward to receiving your revised manuscript.

Kind regards,

Sameer Sheshrao Gajghate, PhD

Academic Editor

PLOS ONE

3. In the online submission form, you indicated that [Data will be made available on request].

4. We note that Figure 1 in your submission contain copyrighted images. All PLOS content is published under the Creative Commons Attribution License (CC BY 4.0), which means that the manuscript, images, and Supporting Information files will be freely available online, and any third party is permitted to access, download, copy, distribute, and use these materials in any way, even commercially, with proper attribution. For more information, see our copyright guidelines: http://journals.plos.org/plosone/s/licenses-and-copyright.

Reviewers' comments:

Reviewer's Responses to Questions

**Comments to the Author**

1. Is the manuscript technically sound, and do the data support the conclusions?

Reviewer #1: Partly

Reviewer #2: Yes

2. Has the statistical analysis been performed appropriately and rigorously? 

Reviewer #1: N/A

Reviewer #2: Yes

3. Have the authors made all data underlying the findings in their manuscript fully available?

Reviewer #1: Yes

Reviewer #2: Yes

4. Is the manuscript presented in an intelligible fashion and written in standard English?

Reviewer #1: No

Reviewer #2: Yes

5. Review Comments to the Author

Reviewer #1: The authors here present a multi-objective optimization of a shell-and-tube heat exchanger, where the objective is to optimize thermal resistance, transmittance, and efficiency. Starting from 16 experiments they build up regression models by using regression analysis and artificial neural network. The optimization is then done by using the genetic algorithm, showing optimimum solutions by means of a 3D Pareto front.

Even if substantial changes are needed, the topic shown, say using artificial intelligence to predict and optimize heat exchangers, is novel and interesting, so the present paper can be considered for publication if the attached points are addressed by the authors

• The authors essentially use experiments to set up their predictive models. Therefore, they should improve this part by including details like heat losses analysis from the heat exchanger, that doesn't seem to be properly insulated in Fig. 1, and uncertainty analysis

• The authors introduce the heat exchanger equation (Eq. 1). Are they making references to a tube-in-tube heat exchanger, or did they need to use a F correction factor to account for the shell-and-tube geometry?

• In subsection 3.2.2, the obtained R2 generally look quite weak for an accurate prediction. Why didn't the authors consider multiple non-linear regression analysis to improve R2? Besides, why the regression looks so weak for the variable TR when compared to the other variables (E and U) shown in subsections 3.2.1 and 3.2.3?

• The authors claim that they use 16 samples for the regression analysis, including the neural network generation (see subsection 3.3 and Fig. 6). Aren't these too few? This would mean that they just use about 3-4 samples for validation and testing

• In Fig. 7 and similar, please use different labels for the figures, since taget and output are too generic. Generally speaking graphics must be improved in this paper

• The authors generally compare experiments, response surface method, and neural network (see for instance Table 6). Since this might be a novelty here, it is suggested to show a dedicated section that compare pro/cons of each approach, highlighting computational costs and effort for each of the methods. Of course with references to experiments one might underline the fact that these are by definition costly and long to perform. Similar papers have been recently published about optimization analysis under different regression approaches (doi.org/10.3390/en15197352) so it is suggested to compare the present outcomes with the ones already available in literature

• When presenting Fig. 13, the authors should avoid negative variables even if they minimize the negative of a variable. Besides, even if it is not that easy in a 3D space, they should try to show dominated points

• Genetic algorithm is generally helpful if one wants to reduce the number of computations to achieve optimum. Therefore, could the authors provide more information about computational time and costs to underline the reason why the genetic algorithm is here necessary? Or did they just assume that since the experimental campaign is generally long then the overall optimization procedure is time costly and using the genetic algorithm might be of help?

Reviewer #2: Thank you for the opportunity to review this manuscript. I have evaluated the manuscript based on the provided questions and have found it to be technically sound, with appropriate statistical analysis and fully available data. The presentation is clear . However, I would like to suggest a few improvements: In the introduction, please clarify the specific challenge addressed in thermal engineering. Elaborate on practical implications and applications in the discussion section. Simplify language to improve readability. Add transitional phrases for coherence. Addressing these points will enhance the clarity, coherence, and accessibility of the paper. Overall, the manuscript meets the standards for publication in PLOS ONE. The authors are commended for their thorough literature review, novel approach to heat exchanger design optimization, and clear explanation of methodology. The paper's practical applicability is demonstrated through numerical examples and case studies, addressing key aspects like energy efficiency, cost minimization, and environmental sustainability. The manuscript is well-structured, and the authors have ensured accuracy and validity through rigorous testing and statistical analysis. This valuable contribution to thermal engineering will benefit the scientific community and has the potential to make a significant impact in the field.

6. PLOS authors have the option to publish the peer review history of their article (what does this mean?). If published, this will include your full peer review and any attached files.

Reviewer #1: No

Reviewer #2: **Yes: **Vibhu Sharma

---

## [Author Response · Author response to Decision Letter 0]

3 Apr 2024

Original Manuscript ID: PONE-D-24-06376 

Original Article Title: “Performance optimization for an optimal operating condition for a Shell and Heat Exchanger Using a Multi-Objective Genetic Algorithm Approach”

To: PloS One Editor

Re: Response to reviewers

Dear Editor,

Thank you for allowing a resubmission of our manuscript, with an opportunity to address the reviewers’ comments.

We are uploading (a) our point-by-point response to the comments (below) (response to reviewers), (b) an updated manuscript with highlighting indicating changes.

Best regards,

Mudassir Khan et al.

---

## [Decision Letter · Decision Letter 1]

7 May 2024

Performance optimization for an optimal operating condition for a Shell and Heat Exchanger Using a Multi-Objective Genetic Algorithm Approach

PONE-D-24-06376R1

Dear Dr. Mudassir Khan,

We’re pleased to inform you that your manuscript has been judged scientifically suitable for publication and will be formally accepted for publication once it meets all outstanding technical requirements.

Kind regards,

Sameer Sheshrao Gajghate, PhD

Academic Editor

PLOS ONE

Additional Editor Comments (optional):

Reviewers' comments:

Reviewer's Responses to Questions

**Comments to the Author**

1. If the authors have adequately addressed your comments raised in a previous round of review and you feel that this manuscript is now acceptable for publication, you may indicate that here to bypass the “Comments to the Author” section, enter your conflict of interest statement in the “Confidential to Editor” section, and submit your "Accept" recommendation.

Reviewer #1: All comments have been addressed

Reviewer #2: All comments have been addressed

2. Is the manuscript technically sound, and do the data support the conclusions?

Reviewer #1: Yes

Reviewer #2: Yes

3. Has the statistical analysis been performed appropriately and rigorously? 

Reviewer #1: (No Response)

Reviewer #2: Yes

4. Have the authors made all data underlying the findings in their manuscript fully available?

Reviewer #1: Yes

Reviewer #2: Yes

5. Is the manuscript presented in an intelligible fashion and written in standard English?

Reviewer #1: Yes

Reviewer #2: Yes

6. Review Comments to the Author

Reviewer #1: The paper can be accepted now. The reviewers replied to everything, with particular references to both experiments and statistical analysis description.

Reviewer #2: This well-structured manuscript presents an innovative approach to optimizing shell and heat exchangers using a multi-objective genetic algorithm. The authors effectively integrate Principal Component Analysis, Response Surface Methodology, Neural Network Fitting, and Genetic Algorithms to improve energy efficiency and equipment lifespan. The experimental setup and methodology are clearly explained, and the results show promising outcomes. Overall, a valuable contribution to the field, well-written and easy to follow.

7. PLOS authors have the option to publish the peer review history of their article (what does this mean?). If published, this will include your full peer review and any attached files.

Reviewer #1: No

Reviewer #2: **Yes: **Vibhu Sharma

---

## [Editor Report · Acceptance letter]

30 May 2024

PONE-D-24-06376R1 

PLOS ONE

Dear Dr. Khan, 

I'm pleased to inform you that your manuscript has been deemed suitable for publication in PLOS ONE. Congratulations! Your manuscript is now being handed over to our production team.

Kind regards, 

on behalf of

Dr. Sameer Sheshrao Gajghate 

Academic Editor

PLOS ONE